# Genome-Wide Identification and Characterization of the Sweet Orange (*Citrus sinensis*) GATA Family Reveals a Role for CsGATA12 as a Regulator of Citrus Bacterial Canker Resistance

**DOI:** 10.3390/ijms25052924

**Published:** 2024-03-02

**Authors:** Jie Fan, Baohang Xian, Xin Huang, Qiyuan Yu, Miao Zhang, Chenxi Zhang, Ruirui Jia, Shanchun Chen, Yongrui He, Qiang Li

**Affiliations:** 1Integrative Science Center of Germplasm Creation in Western China (CHONGQING) Science City, Citrus Research Institute, Southwest University, Chongqing 400712, China; 2National Citrus Engineering Research Center, Chongqing 400712, China

**Keywords:** citrus bacterial canker (CBC), *Xanthomonas citri* subsp. *citri* (*Xcc*), GATA transcription factor, reactive oxygen species (ROS), abscisic acid (ABA), jasmonic acid (JA), salicylic acid (SA)

## Abstract

Citrus bacterial canker (CBC) is a severe bacterial infection caused by *Xanthomonas citri* subsp. *citri* (*Xcc*), which continues to adversely impact citrus production worldwide. Members of the GATA family are important regulators of plant development and regulate plant responses to particular stressors. This report aimed to systematically elucidate the *Citrus sinensis* genome to identify and annotate genes that encode GATAs and evaluate the functional importance of these CsGATAs as regulators of CBC resistance. In total, 24 CsGATAs were identified and classified into four subfamilies. Furthermore, the phylogenetic relationships, chromosomal locations, collinear relationships, gene structures, and conserved domains for each of these GATA family members were also evaluated. It was observed that *Xcc* infection induced some CsGATAs, among which CsGATA12 was chosen for further functional validation. CsGATA12 was found to be localized in the nucleus and was differentially upregulated in the CBC-resistant and CBC-sensitive Kumquat and Wanjincheng citrus varieties. When transiently overexpressed, CsGATA12 significantly reduced CBC resistance with a corresponding increase in abscisic acid, jasmonic acid, and antioxidant enzyme levels. These alterations were consistent with lower levels of salicylic acid, ethylene, and reactive oxygen species. Moreover, the bacteria-induced CsGATA12 gene silencing yielded the opposite phenotypic outcomes. This investigation highlights the important role of CsGATA12 in regulating CBC resistance, underscoring its potential utility as a target for breeding citrus varieties with superior phytopathogen resistance.

## 1. Introduction

The GATA family members are transcriptional regulatory proteins and are named after their ability to bind to W-GATA-R (W = T/A, R = G/A) sequences in the target gene’s promoter region, thereby regulating growth, development, and metabolic activity in plants [1]. These proteins contain DNA-binding domains with a class IV zinc finger structure (C-X_2_-C-X_17–20_-C-X_2_-C) and other basic regions [2]. Based on the conserved exon–intron gene structures and amino acid sequences, GATA family proteins can be classified into four subgroups [3]. Furthermore, based on the conserved domains, *Arabidopsis thaliana* B-GATA proteins can be further subclassified into the N-terminal HAN domain and C-terminal LLM domain [4]. To date, many *GATA* gene family members have been classified by systematic annotation, such as 30, 28, 64, 28, 30, 96, 31, 49, and 79 members of the GATA members have been annotated in Arabidopsis [5], rice [6], soybean [7], millet [3], tomato [1], rapeseed [8], *Phyllostachys pubescens* [9], potato [10], and wheat [11], respectively. These gene families offer a valuable reference to further explore the biological characteristics and functions of members of this GATA transcription factor family.

Recent research has increasingly clarified the functions of GATAs in different plant species, with evidence demonstrating their ability to regulate growth, development of specific tissues, chlorophyll biosynthesis, chloroplast development, stomatal formation, and photosynthetic activity [12]. For example, in *A. thaliana,* the GATA proteins AtGNC and AtGNL can control chloroplast formation and chlorophyll synthesis in lightly growing Arabidopsis seedlings [13]. Similarly, in soybean (*Glycine max*), GmGATA58 functions as a key regulator of chlorophyll biosynthesis [14]. Moreover, the PdGATA19/PdGNC member of this protein family can activate PdNRT2.4b, the high-affinity nitrate transporter gene promoter, which increases nitrate uptake when its levels are reduced to enhance growth and nitrogen use efficiency [15]. PbGATAs in pear (*Pyrus bretschneideri*) plants reportedly modulate hormone signaling pathway activity [16], and the wheat TaGATA1 protein can control TaABI5 expression to promote wheat seed dormancy, thus enhancing resistance to pre-harvest sprouting [17].

Members of the GATA family also play a crucial role in maintaining plant homeostasis under stressful conditions. For instance, when nitrogen concentrations are low, GNC levels in *A. thaliana* increase by 1.5-fold relative to high nitrogen conditions [18]. In soybean seedlings, GATA44 and GATA58 expression levels decline by 81% and 79% relative to control plants, indicating the sensitivity of these GATAs to nitrogen stress [7]. Furthermore, in rice seedlings, OsGATA16 has been shown to help augment cold tolerance by activating genes such as *OsCYL4*, *OsSRFP1*, *OsWRKY45-1*, and *OsMYB30* [19]. After *Vigna subterranea* was exposed to low-temperature stress, the expression of GATA9 reduced, while that of associated downstream genes was upregulated [20]. An RNA sequencing-based analysis of *Brassica juncea* identified 29 GATA family members, 5 and 2 of which were downregulated in response to drought and high temperature, indicating that these proteins regulate abiotic stress responses in plants [21].

Citrus bacterial canker (CBC) is an important bacterial disease caused by *Xanthomonas citri* subsp. *Citri* (*Xcc*) poses a threat to citrus crops worldwide [22]. CBC seriously restricts the development of the citrus industry, and most commercially grown citrus varieties are highly susceptible [23]. CBC occurs on the stems, leaves, and fruits of citrus, and in severe cases, it can cause leaf loss, fruit loss, and even death [22,24]. Many studies have indicated a close association between GATA family members and plant growth, development, and abiotic and biotic stress responses. In this investigation, the *C. sinensis* GATA family was systematically evaluated by elucidating gene structures, phylogenetic relationships, and responses to *Xcc* infection, using susceptible variety Wanjincheng and resistant variety Kumquat as the experimental materials. Moreover, the overexpression and knockdown studies validated that CsGATA12 is a regulator of CBC resistance. These experiments revealed that CsGATA12 regulates sweet orange responses to *Xcc* infection by modulating the biosynthesis of phytohormones and associated signaling pathways, as well as reactive oxygen species (ROS) homeostasis. This might highlight the significant roles played by CsGATA12 in the development of CBC and suggest that CsGATA12 may be of value as a target for the breeding of CBC-resistant citrus plants.

## 2. Results

### 2.1. Systematic Analyses Identified 24 CsGATAs in the C. sinensis Genome

To identify all putative *C. sinensis*-encoded *GATA* genes, 30 known *A. thaliana* encoded GATA proteins were used as queries to search the annotated *C. sinensis* genome in CPBD. The results were manually reviewed, which identified 24 putative CsGATAs that were numbered according to the chromosomal locations of the associated genes (Table 1, Appendix A and Appendix A). The predicted proteins encoded by these *CsGATA* genes ranged from 135 (CsGATA16) to 542 (CsGATA18) amino acids in length, with molecular weights from 14,885 to 60,299 Da. Of these 24 proteins, 14 were overall basic (pI > 7) as they primarily comprised basic amino acids, whereas the remaining 10 CsGATAs were predicted to be acidic (pI < 7) based on amino acid content. CELLO predictions suggested that all 24 of these CsGATAs were localized in the nucleus (Appendix A).

### 2.2. CsGATA Family Chromosomal Locations and Phylogenetic Relationships

To evaluate the phylogenetic relationships among CsGATAs, the protein sequences for these 24 transcription factors were compared with those of 30 AtGATAs (Appendix A). The resultant phylogenetic tree incorporated all 54 of these proteins and classified these GATAs into four major clusters (Figure 1A). However, the representation of these proteins was not balanced within a given cluster. For example, cluster 1 contained 11 CsGATAs and 15 AtGATAs, whereas cluster 2 contained 8 CsGATAs and 11 AtGATAs. Furthermore, CsGATAs were also unbalanced across clusters, with 11, 8, 4, and 1 of these proteins present in clusters 1, 2, 3, and 4, respectively (Figure 1A). Moreover, these 24 *CsGATAs* were also unevenly distributed across the *C. sinensis* genome, being encoded on eight chromosomes and absent from chromosome 6 (Figure 1B). According to the phylogenetic and chromosomal localization results, there were five instances of tandem CsGATA duplication (*CsGATA01*/*02*, *CsGATA05*/*06*, *CsGATA07*/*08*, *CsGATA09*/*10*, *CsGATA12*/*13*), one instance of segmental duplication (*CsGATA13*/*14*), and two instances of whole genome duplications (*CsGATA04*/*11*, *CsGATA10*/*21*), contributing to the expansion of this transcription factor family (Figure 1B).

### 2.3. Analyses of CsGATA Collinearity, Conserved Motifs, and Gene Structures

To better understand the phylogeny of the *CsGATA* gene family, a comparative syntenic map was constructed using GATAs encoded by *C. sinensis* and *A. thaliana,* which revealed a syntenic relationship between eight *CsGATA* genes and members of the *AtGATA* gene family (Figure 2A). These homologous genes may have served important functional roles in the evolution of this gene family. Conserved motifs and gene structures were analyzed to identify the *GATA* gene family and provide the basic characteristics of the GATA family. The motif analyses were performed using MEME, which revealed several motifs, including one (Motif 1) that was conserved across all 24 CsGATAs (Figure 2B). Furthermore, GSDS was used to visualize *CsGATA* gene exon–intron structures, which indicated significant variations in the numbers of exons per gene, ranging from 1 to 11 (Figure 2B). Similar gene structures, however, were observed for closely related CsGATA pairs (*CsGATA01*/*02*, *CsGATA07*/*08*, *CsGATA09*/*10*, and *CsGATA12*/*13*) (Figure 2B).

### 2.4. CsGATA12 Is an Xcc-Inducible Protein Localized in the Nucleus

RNA-seq analyses were conducted to identify seven CsGATAs among the DEGs expressed in response to *Xcc* infection (Appendix A). Both Wanjincheng and Kumquat varieties indicated CsGATA04 downregulation during infection, except at 12 hpi when it was upregulated in Wanjincheng plants. CsGATA13 and CsGATA20 also exhibited comparable expression patterns in both citrus varieties, transitioning from upregulation to downregulation with time. Furthermore, CsGATA14, CsGATA15, and CsGATA17 indicated marked downregulation in response to *Xcc* in both Wanjincheng and Kumquat plants. Although CsGATA12 was downregulated by *Xcc* in both varieties compared with the beginning of induction, the downregulation trends of CsGATA12 were significantly different in the two species. It was progressively downregulated in Kumquat plants, whereas the degree of downregulation was significantly reversed after 12 hpi in Wanjincheng. These data indicated that different CsGATAs respond differentially to *Xcc* infection, suggesting potential roles for these transcription factors as regulators of CBC susceptibility or resistance. Subsequently, CsGATA12 was selected for the functional validation of GATA and the elucidation of its regulatory roles. The qPCR results of changes in *CsGATA12* expression after *Xcc* infection were consistent with RNA-seq analyses (Figure 3A). To confirm the predicted nuclear localization of CsGATA12, it was tagged with GFP and transiently expressed in *A. thaliana* protoplasts. This assay confirmed that the green signal in these protoplasts was exclusively localized to the nuclear compartment, whereas in control protoplasts, green fluorescence was detected in both the nucleus and the cytoplasm (Figure 3B).

### 2.5. Transiently Overexpressing CsGATA12 Sensitizes Wanjincheng Plants to CBC

To test the effects of transient overexpression of this gene, a CsGATA12 overexpression plasmid construct was prepared (Figure 4A). This plasmid successfully upregulated CsGATA12 in Wanjincheng plants, as confirmed by qPCR (Figure 4B). Upon *Xcc* infection, plants overexpressing CsGATA12 exhibited more severe symptoms of CBC than the control plants (Figure 4C), with 1.31-fold larger lesion size (Figure 4D) and 1.25-fold higher disease index values than the control values (Figure 4E). Phytohormones, including ethylene (ET), JA, salicylic acid (SA), and abscisic acid (ABA), are important regulators of plant immunity [25,26,27]. Compared with wild-type (WT) leaves, the leaves of plants overexpressing CsGATA12 exhibited elevated JA levels but reduced SA and ET levels (Figure 4F–I). Furthermore, phytopathogen responses are often dependent on ROS overproduction and the consequent oxidative damage to pathogens or apoptotic death of plant cells [28]. These transgenic cells also exhibited reduced $O_{2}^{-}$ and H_2_O_2_ concentrations (Figure 4J,K). Additionally, the examination of catalase (CAT) and superoxide dismutase (SOD) expressions, responsible for the maintenance of ROS homeostasis, revealed that the activity levels for both enzymes were enhanced by CsGATA12 overexpression, potentially accounting for the reduced ROS levels (Figure 4L,M). Together, these results provide preliminary evidence suggesting that CsGATA12 influences CBC resistance by modulating phytohormone biosynthesis and ROS homeostasis.

### 2.6. VIGS-Mediated CsGATA12 Knockdown Enhances CBC Resistance

A VIGS approach was used to knock down CsGATA12 in Wanjincheng plants to better understand the function of this gene in regulating the resistance to CBC. The TRV2 vector was used to construct a VIGS plasmid (Figure 5A), and with the help of PCR, the success of the VIGS approach was confirmed (Figure 5B) as a significant reduction was observed in CsGATA12 expression compared to the control plants (Figure 5C). Then, the responses of these VIGS plants to *Xcc* inoculation were assessed, which revealed that CBC symptom severity was significantly abrogated by the silencing of this gene (Figure 5D). CsGATA12 knockdown also reduced CBC lesion size to 48.0% of the control lesions (Figure 5E), while disease index values fell to 43.5% of control values (Figure 5F). This suggests that CsGATA12 silencing can enhance CBC susceptibility. These data, together with the CsGATA12 overexpression results, support the classification of *CsGATA12* as a CBC susceptibility gene.

### 2.7. CsGATA12 Silencing Alters Phytohormones and ROS Levels

Subsequently, the phytohormone and ROS levels were also evaluated in these CsGATA12-VIGS plants, which indicated that *CsGATA12* gene silencing enhanced the SA and ET levels in these plants, whereas ABA and JA levels were reduced (Figure 6A–D). CsGATA12 silencing also increased $O_{2}^{-}$ and H_2_O_2_ levels while suppressing CAT and SOD activity (Figure 6E–H). These results indicated that CsGATA12 knockdown augmented resistance to CBC in part by enhancing the levels of SA, JA, and ROS in sweet orange plants.

### 2.8. The Silencing of CsGATA12 Alters the Expression of Phytohormone Biosynthesis and Signaling-Related Genes 

The effects of CsGATA12 silencing on the expression of genes related to ABA, SA, JA, and ET biosynthesis and signaling were assessed [29,30,31,32,33]. It was observed that CsGATA12 knockdown downregulated ABA biosynthesis (*CsNCED1-1* and *CsNCED1-2*) and signaling (*CsPP2C-3* and *CsPP2C-8*) genes (Figure 7A–D), while upregulated the ET biosynthesis gene *CsACS6* (Figure 7E). The upregulation of the *CsPAL1-1* and *CsICS* genes involved in SA synthesis was detected after CsGATA12 silencing, whereas the related *CsNPR1* gene remained unaffected. Moreover, the SA signaling gene, *CsWRKY70*, was also upregulated in these plants (Figure 7F–J). Additionally, the downregulation of genes involved in JA biosynthesis (*CsLOX2*, *CsAOS1-1*, *CsAOS1-2*, *CsOPR3*) and signaling (*CsMYC2-1*) was also observed (Figure 7K–O). Overall, these data support that CsGATA12 is a regulator of phytohormone production and signaling in sweet orange plants, thereby regulating the resistance of plants against CBC.

## 3. Discussion

In this investigation, the *C. sinensis* GATA protein family was systematically evaluated to explore the characteristics and expression patterns of these transcription factors to evaluate their importance as regulators of CBC resistance. A total of 24 CsGATAs encoded in the *C. sinensis* genome were identified, suggesting that sweet orange plants encode fewer GATAs than other plant species [1,3,5,6,7,8]. A lot of research has been conducted on GATAs as regulators of growth, development, and responses to abiotic stressors; however, far less is known about their influence on biotic stress responses. In wheat, TaGATA1 has been characterized as a transcriptional activator that can positively regulate sheath blight defense responses by promoting the upregulation of specific JA signaling-mediated defensive genes [4]. Here, the evaluation of the expression dynamics of CsGATAs in response to *Xcc* infection identified seven CsGATAs, which were potentially associated with CBC resistance and susceptibility (Appendix A). Of these, CsGATA12 was selected for further functional validation as it was differentially induced in CBC-susceptible and CBC-resistant citrus plants. Subsequent transient overexpression and VIGS-mediated knockdown experiments confirmed that CsGATA12 acted as a CBC susceptibility-associated factor (Figure 4 and Figure 5).

Furthermore, JA, SA, ABA, and other phytohormones are important regulators of the plant’s immunity-related signaling activity [25,26,27]. In *Arabidopsis*, JA can mediate immunity against necrotrophic pathogens, and SA plays a crucial role in immunity against pathogens with a biological nutritional lifestyle [34]. In plants, SA does not function independently and interacts with other phytohormones pathways to regulate plant resistance. Phytohormones ABA can also interact with other phytohormones, such as JA and SA. Regulating the ABA signal can antagonize SA-mediated defense. If the ABA level increases, it can increase the content of JA, leading to a decrease in SA [35,36]. It has been reported that ABA and JA are negative regulators of resistance to CBC [37], whereas SA and ET can augment resistance against this pathogen [37,38]. Therefore, to characterize the mechanistic role of CsGATA12 on this basis, its association with hormone signaling and biosynthesis pathways was evaluated, which revealed that it can positively regulate JA and ABA signaling while suppressing SA and ET synthesis and signaling (Figure 4, Figure 6 and Figure 7). GATA proteins are also reportedly essential determinants of a plant’s ability to mitigate biological stressors. In wheat, for example, TaGATA1 functions as a positive regulator of defense responses induced by *Rhizoctonia cerealis* through its ability to bind and activate specific jasmonic acid (JA) signaling-mediated defensive genes [4].

Furthermore, in addition to phytohormones, $O_{2}^{-}$, H_2_O_2_, and other ROS can serve as key signaling intermediates that modulate plant immune responses, such as those that govern CBC resistance [39]. At the same time, CAT also plays a crucial role in plant disease stress. In the early stages of disease occurrence, the activity of CAT is inhibited by salicylic acid (SA), which increases the H_2_O_2_ content in the plant body and makes the plant respond to stress. But, as the content of H_2_O_2_ continues to increase, it will cause phenylpropane to generate SA, triggering SAR [40], effectively resisting bacterial infection [41]. Overall, the impact of CsGATA12 on ROS levels and the activity of SOD and CAT was evaluated. These data indicated that GATA family members regulate CBC resistance by modulating ROS contents (Figure 4 and Figure 6).

Furthermore, the above data support that CsGATA12 can sensitize citrus plants to CBC because of its ability to modulate phytohormone production, signaling, and ROS homeostasis. The results from CBC-susceptible and CBC-resistant citrus varieties and functional data were used to construct a hypothetical model explaining the acquisition of greater CBC resistance in Kumquat and CsGATA12-VIGS plants. Compared with Kumquat plants, the *Xcc* infection of Wanjincheng plants did not significantly downregulate CsGATA12, thereby reducing SA and ET levels while elevating levels of ABA and JA as well as ROS scavenging activity, thus causing increased CBC susceptibility. In contrast, Kumquat plant’s *Xcc* infection promotes CsGATA12 downregulation, thereby increasing the levels of SA, ET, and ROS, causing increased CBC resistance.

## 4. Materials and Methods

### 4.1. Plant and Bacterial Materials

The Kumquat (*Fortunella japonica*) and Wanjincheng (*C. sinensis*) citrus varieties were grown in the National Citrus Germplasm Resource Bank in Chongqing, China (coordinates 19°51′ N, 106°37′ E). These plants were raised in a 28 °C greenhouse. Both the varieties were infected with *Xcc* for subsequent analysis. For transient overexpression and VIGS analyses, only the Wanjincheng variety was utilized. The XccYN1 *Xcc* variety was obtained from the leaves of CBC-susceptible citrus plants and routinely cultured at 28 °C in peptone–yeast extract/malt extract containing 1.5% (*w*/*v*) D-glucose.

### 4.2. CsGATA Family Identification and Annotation

*C. sinensis* genomic and proteomic datasets were downloaded from the Citrus Pan-Genome to Breeding Database (CPBD) [42,43,44] “http://citrus.hzau.edu.cn (accessed on 20 October 2022)” and Phytozome v12 [45] “https://phytozome.jgi.doe.gov/pz/portal.html (accessed on 20 October 2022)”. Then, the CsGATA genes were identified using Hidden Markov Model (HMM) profiles of the GATA family with default parameters [42,43,44] and a 0.01 cut-off value. Additionally, candidate CsGATAs with conserved GATA domains were further analyzed with SMART [46] “http://smart.embl-heidelberg.de (accessed on 11 November 2022)”. The NCBI’s conserved domain database “http://www.omicsclass.com/article/310 (accessed on 11 October 2022)” and CitGVD v1.0 (Citrus Genomic Variations Database) [47] “http://citgvd.cric.cn/home (accessed on 11 October 2022)” were subsequently used to validate the status of these genes as members of the GATA family. Furthermore, ProtParam “http://web.expasy.org/protparam (accessed on 12 November 2022)” was used to assess molecular weight (MW) and isoelectric point (pI) values for these CsGATAs [48]. Additionally, for the predictions of subcellular localization, CELLO V2.5 [49] “http://cello.life.nctu.edu.tw (accessed on 12 November 2022)” was employed.

### 4.3. In Silico CsGATA Characterization

MEGA XI [50] was used to construct a maximum-likelihood (ML) phylogenetic tree based on aligned full-length GATA protein sequences (Appendix A) with 500 bootstrap replicates [51]. The chromosomal locations of individual *CsGATA* genes were visualized via MapChart v2.1 [52]. TBtools v1.098 was used to evaluate the collinearity between *AtGATAs* and *CsGATAs* [53]. Furthermore, for gene structure assessments, GSDS v2.0 [54] “http://gsds.cbi.pku.edu.cn (accessed on 20 December 2022)” was utilized, while MEME v5.1 [55] “http://meme-suite.org/tools/meme (accessed on 20 December 2022)” was used for conserved motif identification, in which the number of 15 motifs was selected. All qPCR primers were designed using the NCBI Primer-BLAST “https://www.ncbi.nlm.nih.gov/tools/primer-blast (accessed on 20 December 2022)” tool, selection of PCR production scale 70–100, and reference gene selection for *Citrus sinensis*.

### 4.4. Xcc Inoculation

To identify the CBC-related GATAs, responses to *Xcc* infection were analyzed using susceptible variety Wanjincheng and resistant variety Kumquat as the experimental materials. For *Xcc* inoculation experiments, Wanjincheng and Kumquat leaves were treated as reported previously [56]. Briefly, fresh harvested leaves were transferred into a 15 cm petri dish and washed with 75% ethanol and ddH_2_O. *Xcc* was cultured overnight to an OD_600_ of 0.5 and then used to inoculate the prepared leaves. Samples were then harvested at 0, 6, 12, and 24 h post-inoculation (hpi) for downstream analyses.

### 4.5. Subcellular Localization Analyses

The protoplasts of *A. thaliana* were extracted from its leaves via an Arabidopsis Protoplast Preparation and Transformation Kit (Coolaber, Beijing, China). The F_SC-GATA12_/R_SC-GATA12_ primer pair was used to amplify the CsGATA12 ORF (Appendix A), and this sequence was then ligated into the pBI221 vector. Subsequently, the *A. thaliana* protoplasts were transformed with the resultant fusion construct or the empty pBI221 vector. At 18 hpi, protein subcellular localization was evaluated using nucl-mCherry as a nuclear marker under the laser scanning confocal microscope (FV1000 viewer, Olympus, Germany). The excitation wavelengths used for GFP and nucl-mCherry were 540 nm and 600 nm, respectively.

### 4.6. Transient Citrus Transformation Assays

The pGLNe-CsGATA12 overexpression plasmid was prepared by amplifying the full-length CsGATA12 coding sequence (CDS) with the F_OEC-GATA12_/R_OEC-GATA12_ primers and inserting this sequence into the pGLNe vector under the control of the CaMV 35S promoter (Appendix A). Then, these expression vectors were used for transforming *Agrobacterium tumefaciens* EHA105 and cultured until they reached OD_600_ of 0.5. After 2 h of incubation at 28 °C, *Xcc* and overexpressed plasmid-containing *A. tumefaciens* were used to inoculate Wanjincheng leaves. These leaves were subsequently incubated at 28 °C for 5 days, after which samples were collected and used to detect the TRV1, TRV2, and CsGATA12 fragments with the F_OEID-CsGATA12_/R_OEID-CsGATA12_ primers (Appendix A), while CsGATA12 expression was evaluated by qPCR with the F_RT-CsGATA12_/R_RT-CsGATA12_ primers (Appendix A).

### 4.7. Virus-Induced Gene Silencing

VIGS fragments were amplified with the F_VIGS-GATA12_/R_VIGS-GATA12_ primers (Appendix A), which were then inserted into the TRV2 vector to yield the TRV2-CsGATA12 plasmid. Then, for VIGS transformation, *A. tumefaciens* were infiltrated, as reported previously [57]. After 30 days, plants exhibiting green fluorescence were collected, ground, and used for PCR analyses to confirm the presence of TRV1 (F_DEC-TRV1_/R_DEC-TRV1_), TRV2 (F_DEC-TRV2_/R_DEC-TRV2_), and CsGATA12 (F_DEC-TRV2_/R_VIGS-GATA12_) sequences, after which qPCR was performed to evaluate VIGS efficiency (Appendix A).

### 4.8. CBC Resistance Analyses

CBC resistance was assessed in plants after transient VIGS overexpression experiments, as reported previously [58]. Briefly, healthy mature leaves of each plant were selected and then punctured (0.5 mm) at six different places; then, each punctured site was injected with 1 µL of *Xcc*YN1 (1 × 10^8^ CFU∙mL^−1^). Three sets of experiments were repeated. In the transiently overexpression experiments, three leaves were measured in each set, and in VIGS experiments, seven leaves were measured in each set. Leaves were then evaluated for CBC symptoms at 10 days post-inoculation (dpi), with the degree of CBC resistance assessed based on disease lesion area and disease index values [58].

### 4.9. Biochemical Index Measurements

Ultra-high performance liquid chromatography–tandem mass spectrometry (UPLC-MS/MS) was employed for measuring JA, SA, and ABA concentrations, while gas chromatography (GC) was used for assessing the levels of ET, ROS ($O_{2}^{-}$, H_2_O_2_), and antioxidant enzymes (CAT and SOD) using a commercial kit based on provided instructions (Bonoheng, Chongqing, China). Three transient overexpression samples, two VIGS samples were used to measure the biochemical index.

### 4.10. RNA Sequencing

Three Wanjincheng and three Kumquat plants were selected as experimental plant materials, treated with *Xcc* and ddH_2_O, respectively, and RNA sequencing (RNA seq) was performed at 0, 6, 12, and 24 hpi. Each sample’s RNA was extracted via the EASYspin Plant RNA Kit (AidLab, Beijing, China), per the provided directions, after which samples were sent to Majorbio Inc. for sequencing with paired-end sequencing on the Illumina Novaseq 6000 platform. The raw data were filtered by Majorbio cloud platform “https://www.majorbio.com (accessed on 20 May 2023)” to remove the adapter information, low-quality bases, and unmeasured bases to output the clean data. For each sample, about 6G clean data were mapped to the *C. sinensis* genome V3, assembled, and annotated with the CPDB [44]. Subsequently, differentially expressed genes (DEGs) were identified based on the fold-change (FC) in expression when comparing *Xcc*-infected and control plants (FC > 3 or FC < 0.3333). Lastly, GATAs were identified from these DEGs.

### 4.11. qPCR

A kit RN09 used for total RNA extraction (AidLab, China) was used to extract RNA from appropriate samples, after which a PrimeScript kit RR037A (TaKaRa, Beijing, China) was used to process these samples to produce cDNA. Then, using a SYBR Premix kit (Baoguang, Beijing, China) and the Quantagene Real-Time System q225 (Novogene, Beijing, China), qPCR analyses were performed with the following thermocycler settings: 95 °C for 5 min; 40 cycles of 95 °C for 10 s, 56 °C for 30 s. Each 12 μL reaction contained 100 ng of cDNA, 0.5 μM concentrations of appropriate primers, and 6 μL of SYBR Green PCR mix. Each type of sample consists of two groups, each taken from different plants, and each group of experiments is repeated three times technically. Relative expression was quantified via the 2^−∆∆CT^ method [59]. NCBI Primer-BLAST was used to design the CsGATA primers and using CsGAPDH (CPDB ID: Cs_ont_5g044290) [60,61] was used as normalization control and amplified with the F_RT-GAPDH_/R_RT-GAPDH_ primers (Appendix A). Calculate three values for each gene and plot the mean and standard deviation.

### 4.12. Statistical Analysis

All statistical analyses were performed on GraphPad Prism 8.0 (GraphPad, Boston, MA, USA), and the results were compared using two-tailed *t*-tests or ANOVAs with Tukey’s multiple range test. Experiments were conducted in triplicate, and the results were presented as means ± standard deviations (SDs).

## 5. Conclusions

In summary, the present study is the first to systematically evaluate and annotate the *GATA* gene family in *C. sinensis,* enabling the assessment of the evolution and functional significance of these regulatory proteins. This approach ultimately identified 24 CsGATAs with complex patterns of genetic inducibility, suggesting that they play diverse roles in modulating CBC resistance. Transient CsGATA12 overexpression and knockdown revealed that this gene has an important role as a negative regulator of CBC resistance as it reduces SA, ET, and ROS levels in the leaves of sweet orange plants. These data emphasize the important role of CsGATA12 as a regulator of phytopathogen resistance, highlighting its potential as a promising target for breeding citrus plants with higher CBC resistance.

## Figures and Tables

**Figure 1 ijms-25-02924-f001:**
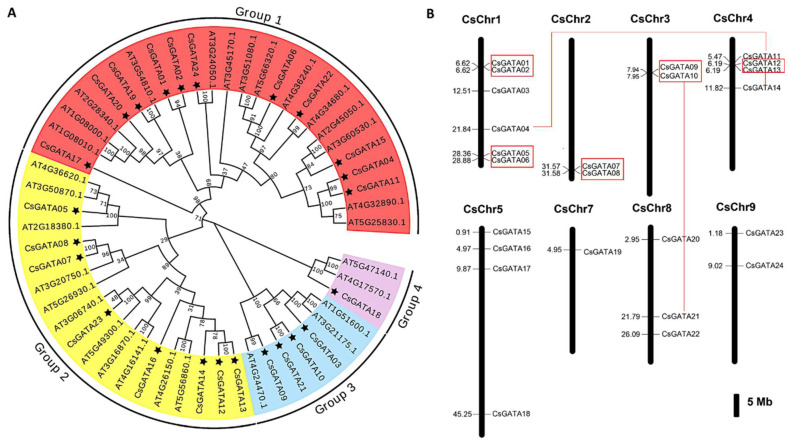
CsGATA phylogenetic relationships and chromosomal locations. (**A**) MEGA XI was used to construct an ML phylogenetic tree comprising GATA family proteins from A. thaliana and *C. sinensis* (AtGATAs and CsGATAs), the latter of which are marked using black stars. The tree was constructed based on complete protein sequences using the Poisson model with 500 bootstrap replicates, and branches are color-coded according to the GATA subclasses. The number of substitutions per site is marked on the branches of the resultant tree. (**B**) The locations of *CsGATA* genes in the *C. sinensis* genome were visualized via MapChart v2.1. The scale indicates the sizes of chromosomes, and black lines denote *CsGATA* gene positions with corresponding numbers indicating their precise locations. Tandem duplication events are indicated using red rectangles, while red lines represent segmental and whole genome duplication.

**Figure 2 ijms-25-02924-f002:**
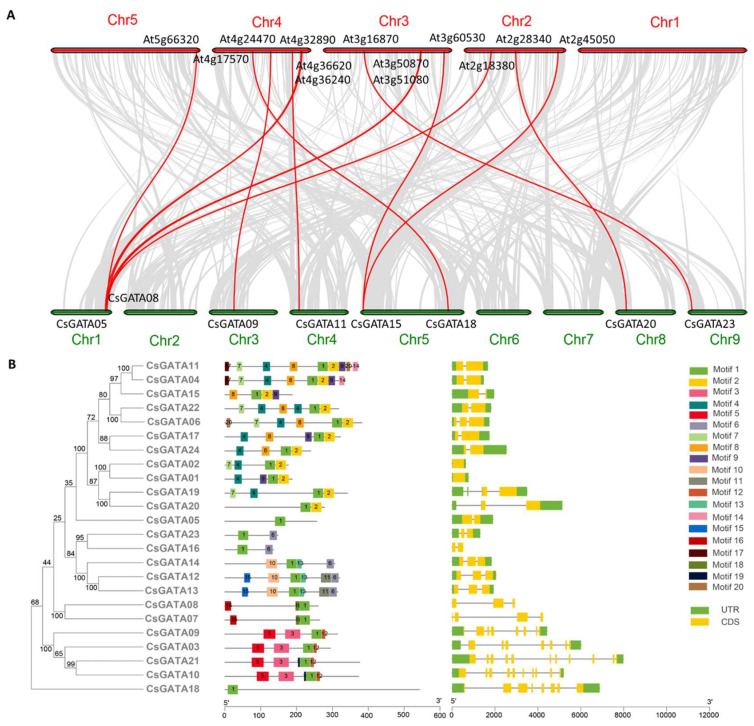
Analyses of CsGATA collinearity, conserved motifs, and gene structures. (**A**) Collinearity among GATAs encoded by *A. thaliana* and *C. sinensis* was visualized. The red line represents the collinearity of *GATA* genes, while the gray lines represent collinearity of genomic segments. (**B**) GSDS v2.0 was used to analyze CsGATA-conserved motifs and gene structures. Introns, exons, and untranslated regions (UTRs) are, respectively, represented with yellow boxes, lines, and green boxes. Intron and exon lengths are indicated with the corresponding scale.

**Figure 3 ijms-25-02924-f003:**
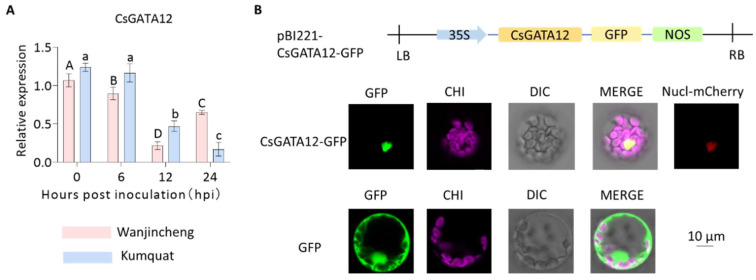
Analyses of the expression and localization of CsGATA12. (**A**) Kumquat and Wanjincheng leaves were infected with *Xcc*, and samples were collected at 0, 6, 12, and 24 hpi for qPCR analysis of CsGATA12 expression using CsGAPDH (CPDB ID: Cs_ont_5g044290) as a normalization control. Different letters indicate significant differences (*p* < 0.05, ANOVAs with Tukey’ s multiple range test). (**B**) The subcellular localization of CsGATA12 was evaluated. GFP, chloroplast luminescence (CH), brightfield (BF), and merged images are shown. Scale bar: 10 μm. 35S, cauliflower mosaic virus 35S promoter; NOS, NOS terminator; GFP, green fluorescent protein; LB: left border; RB: right border.

**Figure 4 ijms-25-02924-f004:**
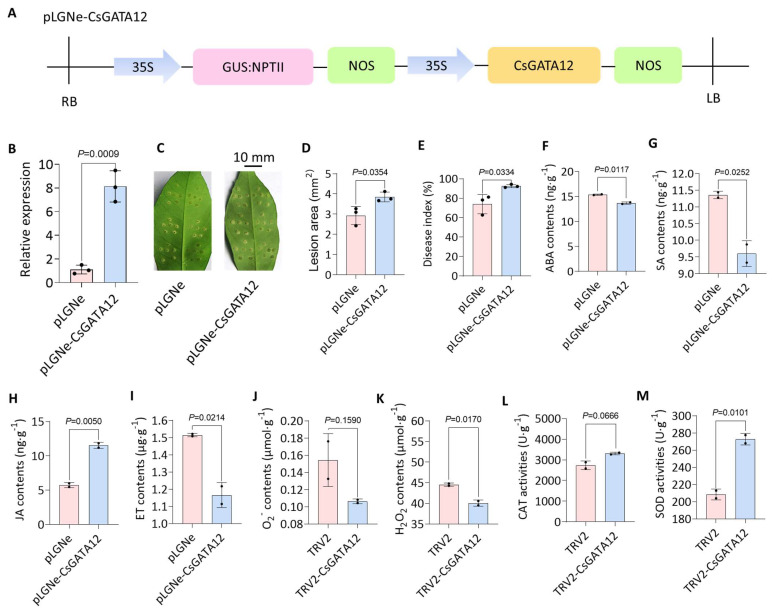
Analyses of the impact of CsGATA12 overexpression on CBC resistance, phytohormone levels, and ROS homeostasis. (**A**) CsGATA12 overexpression vector. 35S, cauliflower mosaic virus 35S promoter; NOS, NOS terminator; GUS: β-glucuronidase; NPTII: β-glucuronidase and NPT integrated coding genes. LB: left border; RB: right border. (**B**) The assessment of CsGATA12 expression following its transient overexpression, where CsGAPDH (CPDB ID: Cs_ont_5g044290) was set as a control for normalization. (**C**) Disease symptoms were evaluated at 10 dpi in *Xcc*-infected leaves overexpressing CsGATA12. Scale bar: 10 mm. (**D**,**E**) CBC lesion sizes (**D**) and disease index values (**E**) were assessed at 10 dpi in plants overexpressing CsGATA12. (**F**–**K**) Using CsGATA12-overexpressing plant samples, ABA (**F**), SA (**G**), JA (**H**), ET (**I**), $O_{2}^{-}$ (**J**), and H_2_O_2_ levels (**K**). (**L**,**M**) The activity levels of CAT (**L**) and SOD (**M**) were assessed in the CsGATA12-overexpressing plant. Results were compared with two-tailed *t*-tests and presented as means ± SDs. pLGNe: Wanjincheng plants expressing the empty control pLGNe vector; pLGNe-CsGATA12: transgenic Wanjincheng plants overexpressing CsGATA12.

**Figure 5 ijms-25-02924-f005:**
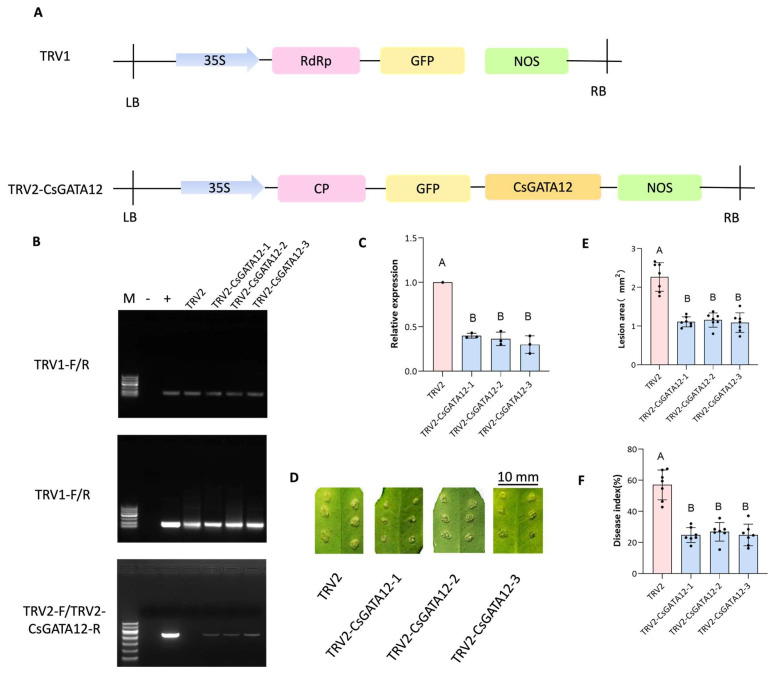
Evaluation of the effect of CsGATA12 silencing on CBC resistance. (**A**) A schematic overview of the VIGS plasmid used in this study. 35S, cauliflower mosaic virus 35S promoter; NOS, NOS terminator; GFP, green fluoresceny protein; CP: coat protein; LB: left border; RB: right border. (**B**) PCR-based confirmation of the successful transformation of VIGS plants. M: DNA ladder. −: negative control (ddH_2_O); +: positive control (plasmid). (**C**) qPCR was used to assess relative CsGATA12 expression, using CsGAPDH (CPDB ID: Cs_ont_5g044290) as a normalization control. (**D**–**F**) VIGS plants were evaluated for *Xcc* infection and then at 10 dpi for disease symptoms (Scale bar: 10 mm) (**D**), lesion sizes (**E**), and disease index values (**F**). In (**C**,**E**–**F**), letters above bars represent the significance of the difference. TRV2-CsGATA12-1, -2, -3: CsGATA12 VIGS plants; TRV2: control plants transformed with the empty TRV2 vector (*p* < 0.05, ANOVAs with Tukey’ s multiple range test).

**Figure 6 ijms-25-02924-f006:**
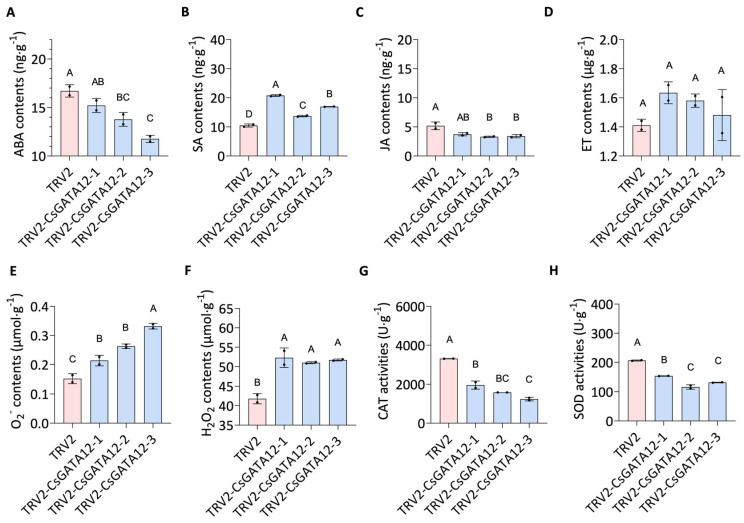
Biochemical analyses of the impact of CsGATA12 knockdown on sweet orange plant phenotypes. The levels of ABA (**A**), SA (**B**), JA (**C**), ET (**D**), $O_{2}^{-}$ (**E**), and H_2_O_2_ (**F**) were analyzed after CsGATA12 knockdown via the VIGS approach. (**G**,**H**) The impact of CsGATA12 silencing on CAT (**G**) and SOD (**H**) activity levels were analyzed. TRV2-CsGATA12-1, -2, -3: CsGATA12 VIGS plants; TRV2: control plants transformed with the empty TRV2 vector (*p* < 0.05, ANOVAs with Tukey’ s multiple range test). Letters above bars represent the significance of the difference.

**Figure 7 ijms-25-02924-f007:**
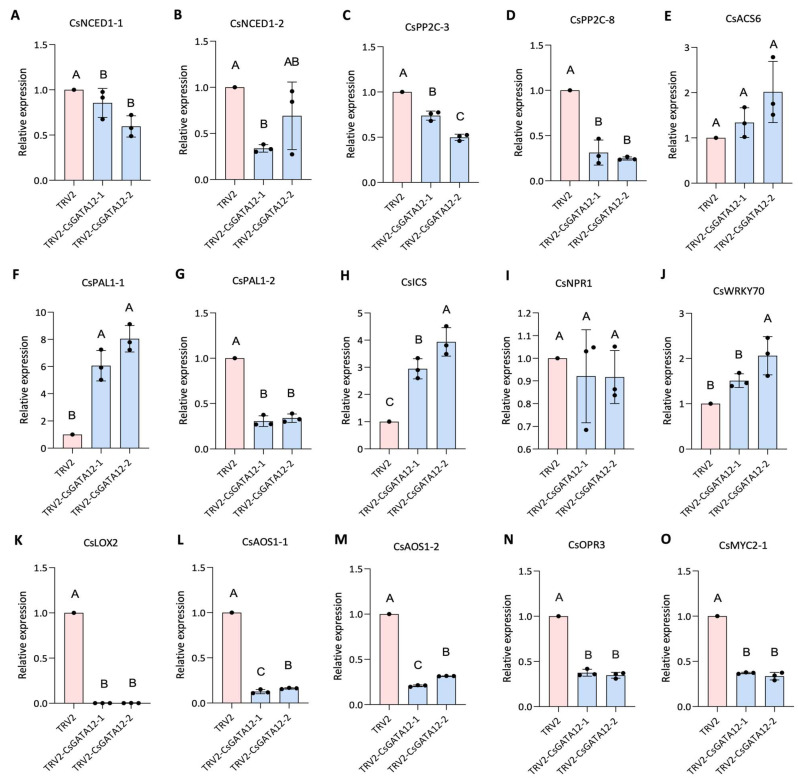
Evaluation of the impact of CsGATA12 silencing on phytohormone biosynthesis and signaling. (**A**–**D**) Relative ABA biosynthesis (**A**,**B**) and signaling (**C**,**D**) gene expression. (**E**) Relative ET biosynthesis-related gene expression. (**F**–**J**) Relative SA biosynthesis (**F**–**H**) and signaling (**I**,**J**) gene expression. (**K**–**O**) Relative JA biosynthesis (**K**–**N**) and signaling (**O**) gene expression. All qPCR analyses were performed using CsGAPDH (CPDB ID: Cs_ont_5g044290) as a reference control. TRV2-CsGATA12-1, -2, -3: CsGATA12 VIGS plants; TRV2: control plants transformed with the empty TRV2 vector (*p* < 0.05, ANOVAs with Tukey’ s multiple range test). Letters above bars represent the significance of the difference.

**Table 1 ijms-25-02924-t001:** Protein characteristics, pI, and subcellular localization of the CsGATA family.

Name	CPBD ID	No. of AA	MW (Da)	pI	Subcellular Loci
CsGATA01	Cs_ont_1g006010.1	187	21,109.18	9.33	Nuclear
CsGATA02	Cs_ont_1g006030.1	176	19,944.61	8.67	Nuclear
CsGATA03	Cs_ont_1g009190.1	293	32,054.5	6.15	Nuclear
CsGATA04	Cs_ont_1g017330.1	334	36,666.63	5.72	Nuclear
CsGATA05	Cs_ont_1g026740.1	255	28,332.15	8.3	Nuclear
CsGATA06	Cs_ont_1g027500.1	381	41,862.23	7.17	Nuclear
CsGATA07	Cs_ont_2g034040.1	263	28,955.71	7.68	Nuclear
CsGATA08	Cs_ont_2g034050.1	259	28,697.63	6.23	Nuclear
CsGATA09	Cs_ont_3g012160.1	313	34,021.14	8.57	Nuclear
CsGATA10	Cs_ont_3g012170.1	372	40,633.73	4.68	Nuclear
CsGATA11	Cs_ont_4g004550.1	372	41,198.54	6.21	Nuclear
CsGATA12	Cs_ont_4g005450.1	319	35,283.36	9.36	Nuclear
CsGATA13	Cs_ont_4g005460.1	314	35,014.26	9.56	Nuclear
CsGATA14	Cs_ont_4g010540.1	306	33,789.07	9.34	Nuclear
CsGATA15	Cs_ont_5g001200.1	187	21,238.65	10.17	Nuclear
CsGATA16	Cs_ont_5g007570.1	135	14,885.36	10.07	Nuclear
CsGATA17	Cs_ont_5g015080.1	321	35,288.43	6.66	Nuclear
CsGATA18	Cs_ont_5g043090.1	542	60,299.05	6.08	Nuclear
CsGATA19	Cs_ont_7g006300.1	341	37,446.96	6.27	Nuclear
CsGATA20	Cs_ont_8g005160.1	277	31,232.25	8.96	Nuclear
CsGATA21	Cs_ont_8g017440.1	375	41,116.14	4.77	Nuclear
CsGATA22	Cs_ont_8g020340.1	316	35,204.05	5.41	Nuclear
CsGATA23	Cs_ont_9g001910.1	147	15,909.97	9.63	Nuclear
CsGATA24	Cs_ont_9g011520.1	238	26,444.29	9.6	Nuclear

All identified *C. sinensis* GATAs are included in this table. ExPASy was used to predict physicochemical properties, whereas for predicting subcellular localization, CELLO v2.5 was employed. CPBD: Citrus Pan-Genome to Breeding Database “http://citrus.hzau.edu.cn (accessed on 20 October 2022)”; AA: amino acid; MW: molecular weight; pI: isoelectric point; Da: dalton.

## Data Availability

The authors confirm that raw RNA-Seq data are archived as Sequence Read Archive (SRA) in the National Center for Biotechnology Information (NCBI) with accession numbers PRJNA913776 and PRJNA913992. Other data supporting the findings of this study are available within the article and its Appendix A.

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
