# Peer review of "Genome-Wide Identification and Characterization of the Sweet Orange (Citrus sinensis) GATA Family Reveals a Role for CsGATA12 as a Regulator of Citrus Bacterial Canker Resistance"

_ijms, 2024, doi:10.3390/ijms25052924_

Round 1

Reviewer 1 Report

Comments and Suggestions for Authors

Genome-wide identification and characterization of the sweet orange (Citrus sinensis) GATA family reveals a role for CsGATA12 as a regulator of citrus bacterial canker resistance

The manuscript describes a two-step approach, first looking at the GATA family as a whole in Citrus sinensis, both at the genomic and transcriptomic levels, secondly focusing on one member of the family, CsGATA12 and its involvement in response to citrus bacterial canker resistance. Thus, a set of 24 genes encoding potential GATA transcription factors are described and CsGATA12 is shown to have a role as a susceptibility factor in the interaction with Xanthomonas citri subsp. Citri.

The manuscript is relatively well written, and its outline is very clear. It involves genome-wide genic and transcriptomic characterization of a transcription factor family, with a wealth of precise softwares, then the functional characterization of the involvement in C. sinensis / Xcc interaction for CsGATA12, again with an array of different evaluations. It represents a large effort and relates to many plant science communities, including both genomics and pathology areas. Thus, this article could have a large public.

However, there are a few points the authors would need to address:

In the already substantial materials and methods section, the authors should add a few details on different points: 1/ in silico CsGATA characterization: On top of the softwares used, the parameters are also an important factor, please ensure they are present so that the analysis can be effectively replicated. 2/ Pathology, transformation and VIGS experiments: the actual experimental design should be given. Indeed, we do not know how many replicates or how many samples are then summarized in the different conclusions. This is also true for the Biochemical index measurements. 3/ RNAseq: what sequencing depth was expected and did the author used paired-ends, of which size?

The authors make an in-depth analysis of the GATA family at the genome level, based on public C. sinensis genomic references and used their own data for a DEG approach. It might have been useful to check within the RNAseq data if the 24 identified GATA encoding genes were all expressed and whether the structural annotation presented in Figure 2 based on in silico analyses were confirmed. This would have been a nice transition towards the analysis in the context of the interaction with Xcc.

Paragraph 3.4 is pivotal in the manuscript as it is the transition from global analysis towards candidates involved in Xcc resistance/susceptibility. A figure illustrating the evolution in expression of the 7 GATA candidates might help clarify why CsGATA12 was selected as the best candidate for the rest of the manuscript. From the text only, I found it difficult, even if Figure 3 shows is pattern of expression. Comparing it to the others might improve the reasoning behind its selection and clarify the comparisons made, whether between constitutive and induced expression post inoculation or between the two resistant and susceptible cultivars.

The evidence for functional validation of CsGATA12 as a susceptibility factor is mainly coming from the accumulation of individual, but sometimes limited (the variations in disease severity variables are just about statistically significant) points. Nevertheless, it is very interesting to see how far one should go to conclude on the effect of a particular gene. As the authors state in the first sentence of the introduction, members of the GATA family can regulate growth, development and metabolic activity in plants. Indeed, the authors show a clear impact of modification of CsGATA12 on different hormones such as ABA, JA and SA. Thus, should this really be a target for efforts to breed citrus plants with superior CBC resistance? What other impact on plant growth and yield formation would a down-regulation of CsGATA12 have? Could the author discuss this point before the conclusion?

Comments on the Quality of English Language

The manuscript might benefit from a complete review by a native english-speaking person. 

Reviewer 2 Report

Comments and Suggestions for Authors

This manuscripts describes genome-wide identification and characterization of GATA family genes in sweet orange (Citrus sinensis). The authors further identified a key GATA gene and its role in bacterial canker resistance in ‘Kumquat’ through transcriptome, overexpression, and gene-silencing experiments. These results proposed an important insight into mechanism of citrus bacterial canker resistance. However, several problems pointed out below should be carefully addressed before publication:

1. Introduction: To make the purpose of this study clearer, the structure should be changed. As the authors exclusively focus on explaining GATA family genes, the study's purpose and background are not clear, as well as being abrupt for the non-specialist reader. A part of detailed description of GATA could be transferred into Discussion and instead, citrus bacterial canker should be first and more described in this section.

2. L100: Why the varieties ‘Kumquat’ and ‘Wanjincheng’ were chosen as plant materials of this study? The reason should be described in Introduction, not only in Discussion (L367-).

3. Method of statistical tests is inappropriate throughout the study. Multiple testing using simple t-test or Duncun’s test makes erroneous inferences (see https://en.wikipedia.org/wiki/Multiple_comparisons_problem). All of the statistical tests must be corrected using an appropriate method like Tukey’s test, including stricter significance threshold.

(Minor points)

1. L82: temperatures, “respectively”.

2. L109: Citations are required for the methods for identifying CsGATA genes.

3. L118: Citation should be added for MEGA XI.

4. L168: Method for preparing ‘Cleaned data’ should be described in detail.

5. L173: Product names should be specified for “A total RNA kit” and “a different kit”.

6. L302: “RdRp” is not shown in Figure 5.

Comments on the Quality of English Language

Generally OK, but occasional mistakes.

Reviewer 3 Report

Comments and Suggestions for Authors

This study is amied to investigate genome-wide identification and characterization of the sweet orange (Citrus sinensis) showing the role for CsGATA12 as a regulator of citrus bacterial canker resistance. The experimental design is acceptable. The study contains some interesting results that can be considered for publication after suitable revisions.

Suggestions:

The study do not follow mpdi format.

L96: At the end of the Introduction, a well defined one sentence objectvies need to be written. 

L181: Give more details on statistical anylses for specific data set.

L196: Title of Table 1 is short and not well informative.

L340: Discussion si too short. Give a more detailed and more deep comparisons of your results with previous studies.

L417: The references section should be adjusted to journal format.

Round 2

Reviewer 2 Report

Comments and Suggestions for Authors

Thank you for revising. I agree that most of my comments have been addressed.

>Response 2:

>To identify the CBC-related GATAs, responses to Xcc infection were >analyzed using susceptible variety Wanjincheng and resistant variety >Kumquat as experimental materials. We have described in Introduction and >the methodology (Lines 88-90, Lines 126-128).

For this comment, I could not comfirm the description in the methodology (Like 126-128). Please reconfirm.

Author Response

Dear reviewer:

We thank you very much for giving us an opportunity to revise our manuscript (Manuscript ID: ijms-2817483). According to your comments, we have revised the manuscript. Revised portions are marked in red in the paper. The main corrections in the paper and the responds to the comments are as follows:

Comment:

To identify the CBC-related GATAs, responses to Xcc infection were analyzed using susceptible variety Wanjincheng and resistant variety Kumquat as experimental materials. We have described in Introduction and the methodology (Lines 88-90, Lines 126-128).

For this comment, I could not confirm the description in the methodology. Please reconfirm.

Response:

We have added and highlighted the descriptions in section Materials and Methods. (Lines 126-128)